# Comparison of Oblique Lumbar Interbody Fusion Combined with Posterior Decompression (OLIF-PD) and Posterior Lumbar Interbody Fusion (PLIF) in the Treatment of Adjacent Segmental Disease(ASD)

**DOI:** 10.3390/jpm13020368

**Published:** 2023-02-19

**Authors:** Bin Zhang, Yuan Hu, Qingquan Kong, Pin Feng, Junlin Liu, Junsong Ma

**Affiliations:** 1Department of Orthopedics Surgery, Hospital of Chengdu Office of People’s Government of Tibetan Autonomous Region, Chengdu 610041, China; 2Department of Orthopedics Surgery, West China Hospital, Sichuan University, Chengdu 610041, China

**Keywords:** oblique lumbar interbody fusion, posterior decompression, posterior lumbar interbody fusion, adjacent segment disease, lumbar spinal stenosis

## Abstract

Background: An unintended consequence following lumbar fusion is the development of adjacent segment disease (ASD). Oblique lumbar interbody fusion combined with posterior decompression (OLIF-PD) is another feasible option for ASD, and there is no literature report on this combined surgical strategy. Methods: A retrospective analysis was performed on 18 ASD patients requiring direct decompression in our hospital between September 2017 and January 2022. Among them, eight patients underwent OLIF-PD revision and ten underwent PLIF revision. There were no significant differences in the baseline data between the two groups. The clinical outcomes and complications were compared between the two groups. Results: The operation time, operative blood loss and postoperative hospital stay in the OLIF-PD group were significantly lower than those in the PLIF group. The VAS of low back pain in the OLIF-PD group was significantly better than that in the PLIF group during the postoperative follow-up. The ODI at the last follow-up in the OLIF-PD group and the PLIF group were significantly relieved compared with those before operation. The excellent and good rate of the modified MacNab standard at the last follow-up was 87.5% in the OLIF-PD group and 70% in the PLIF group. There was a statistically significant difference in the incidence of complications between the two groups. Conclusion: For ASD requiring direct decompression after posterior lumbar fusion, compared with traditional PLIF revision surgery, OLIF-PD has a similar clinical effect, but has a reduced operation time, blood loss, hospital stay and complications. OLIF-PD may be an alternative revision strategy for ASD.

## 1. Introduction

In recent decades, the incidence of lumbar degenerative diseases has gradually increased with the aging of the population [1]. At the same time, with the rapid development of spinal fusion devices, more lumbar fusion operations are used to treat lumbar degenerative diseases and have achieved satisfactory clinical results [2,3]. However, spinal fusion increases the mechanical stress and motion in the adjacent segments, increasing the risk of adjacent segment degeneration [4,5]. Adjacent segment degeneration has been reported in 36–84% of patients five years after lumbar fusion [6]. The degenerative changes of the adjacent segments after lumbar fusion include the loss of the intervertebral disc height, spondylolisthesis, disc herniation, osteophyte formation, facet joint disease and vertebral compression fractures. Adjacent segment disease (ASD), also known as symptomatic adjacent segment degeneration, occurs in approximately 2% to 18.5% [7,8,9]. The main clinical symptoms of ASD are low back pain, radiculopathy, intermittent claudication or cauda equina syndrome. Although the exact pathogenesis of ASD has not been fully clarified, most researchers believe that biomechanical changes are the main cause of ASD. These changes include an increased mobility of the adjacent segments (upper or lower segments) after lumbar fusion, increased pressure in the intervertebral disc and increased load on the facet joints. Conservative treatment is the first choice for ASD with mild clinical symptoms, mainly including bed rest, epidural steroid injection, physical therapy, massage, drug therapy, etc. Revision surgery has become a common option after conservative treatment has failed. The purpose of revision surgery is still the decompression of the neural structures and the reconstruction of spinal stability. The surgical strategy of ASD remains controversial, including extended posterior lumbar interbody fusion (PLIF), anterior lumbar interbody fusion and endoscopic surgery [10]. Among them, extended PLIF is the most common surgical strategy, which has the advantages of direct decompression and a satisfactory clinical effect [11]. However, due to the tissue scarring from the previous operation, the loss of the natural anatomy and the need to deal with the previous hardware, many studies have reported high perioperative complication rates with revision PLIF, making it more challenging in the treatment of ASD [12,13].

The oblique lumbar interbody fusion (OLIF) procedure accesses the intervertebral disc through a natural anatomical approach between the aorta and the psoas muscle. The OLIF technology does not damage the muscles, ligaments and other structures behind the vertebral body, and reduces the risk of postoperative low back pain. OLIF can directly remove a large number of intervertebral disc tissues, so that the fusion cage with a larger contact area can be used, which can greatly increase the support strength of the fusion cage and increase the fusion rate. By avoiding passage through critical anatomical structures, the risk of injury to the blood vessels and nerve plexuses is reduced and the postoperative complications caused by exposure are avoided. In recent years, OLIF has been widely used in various lumbar degenerative diseases that require the reconstruction of intervertebral stability, the restoration of intervertebral space height, indirect decompression and the restoration of the normal lumbar sequence. Compared with traditional intervertebral fusion, OLIF has the advantages of less trauma, less bleeding, faster recovery and a higher fusion rate [14,15]. However, OLIF only provides indirect decompression, which is often insufficient for ASD patients with large disc herniation, ossification of the spinal canal and spinal stenosis [16,17]. In this study, we used oblique lumbar interbody fusion combined with posterior decompression (OLIF-PD) to treat ASD cases requiring direct decompression. To our knowledge, this combined surgical strategy has not been reported in the literature.

## 2. Materials and Methods

### 2.1. Patient Selection

Following IRB approval, we retrospectively reviewed 72 consecutive patients with lumbar ASD at our hospital, who underwent surgery for symptomatic ASD between December 2017 and April 2022. The same experienced spinal minimally invasive surgeon (corresponding author) performed all of the operations. The patients were screened according to the following inclusion and exclusion criteria. The flowchart of the study is shown in Figure 1.

Inclusion criteria: (1) Low back pain, neurogenic claudication or lower extremity radiculopathy after previous posterior lumbar fusion surgery. (2) Instability and spondylolisthesis (≤II°) of adjacent segments found by imaging examination and patients requiring direct decompression, such as obvious herniated disc, central canal stenosis or bony lateral recess stenosis, etc. (3) Aged 18–80 years old. (4) Conservative treatment has been ineffective for more than 3 months.

Exclusion criteria: (1) ASD patients who do not need decompression or only need indirect decompression, such as segmental instability, mild disc herniation, etc. (2) Patients with severe lumbar spondylolisthesis (≥III°), severe osteoporosis. (3) Primary surgery for spinal tumors or infection. (4) Data were incomplete or lost during follow-up.

According to the inclusion and exclusion criteria, 18 ASD patients who underwent revision surgery in our hospital were included. There were 10 males and 8 females. There were 13 previous PLIF operations and 5 transforaminal lumbar interbody fusion operations. The symptoms of the patients were pain in the lower back or lower extremities. All patients underwent lumbar spine MRI, CT, anterior-posterior and lateral X-ray and dynamic X-ray examinations. It was confirmed that the pain was caused by the degeneration of the adjacent segments of lumbar fusion. Among them, 2 cases were L1/2, 5 cases were L2/3, 7 cases of L3/4 and 4 cases of L4/5. After informing them of the advantages and disadvantages of the two types of surgery, the surgical method was selected according to the patient’s wishes, including 8 cases of OLIF-PD and 10 cases of PLIF. There was no significant difference in the baseline data between the two groups (Table 1).

### 2.2. Surgical Procedures

OLIF-PD group: the patient was under general anesthesia, in the right lateral decubitus position. The target intervertebral space was identified by C-arm fluoroscopy and marked on the skin. A 6 cm oblique incision was made 4 cm anterior to the midpoint of the target intervertebral space. The skin, subcutaneous fat and fascia of the external oblique muscle were incised in turn, and the fibers of the external oblique muscle, the internal oblique muscle and the transverse abdominis were bluntly separated. After entering the retroperitoneal space, the fingers are bluntly separated and the retractor retracts the abdominal organs, ureters, blood vessels, etc. The psoas major muscle was retracted posteriorly to expose the target intervertebral disc and insert a probe, and fluoroscopy was performed again to determine the surgical space. After that, the dilators were sequentially implanted, an appropriate retraction baffle was selected, an operating channel was established, and a lighting system was installed. The intervertebral discs were excised, the cartilage endplates were cured and the intervertebral space was placed sequentially using an implant trial to determine the appropriate cage size. The allograft bone and BMP2 (Jiuyuan Gene Engineering Corp., Hangzhou, Zhejiang, China) were filled into the cage and inserted into the intervertebral space, and the position of the cage was confirmed by anteroposterior and lateral fluoroscopy. The screws were implanted on the lateral side of the vertebral body. The screws were close to the middle and lower part of the vertebral body to avoid the posterior pedicle screws. Titanium rods were attached and fixed. The bleeding of the wound was checked again, the operation area was irrigated, the drainage was placed and sutured layer by layer.

Flip to prone position. C-arm fluoroscopy was performed to determine the target intervertebral space, and a 2 cm longitudinal incision was made 2 cm next to the midline. The skin, subcutaneous and fascia were incised layer by layer, the paravertebral muscles were bluntly separated, and a channel with a diameter of 20 mm was placed after gradual expansion. Fluoroscopy is repeated to confirm the segment. The facet joint was exposed and resected, the dural sac and nerve roots were carefully exposed and protected, and the lateral spinal canal and the ventral side of the dural sac were decompressed. The hemorrhage was carefully checked, a drainage tube was placed, and the incision was sutured layer by layer.

PLIF group: general anesthesia, prone position, the patient’s chest and abdomen were suspended. A 15 cm posterior median incision was made, and the skin and deep fascia were incised in layers sequentially. The paraspinal muscles on both sides were peeled off to the inside of the facet joints. The previous internal fixation system was exposed and partially removed. Pedicle screws were placed on both sides of the adjacent segment, and the position of the screws was confirmed by fluoroscopy. Resect the lamina and part of the facet joints of the responsible segment, expose the dural sac and nerve roots, and carefully protect the nerve roots. The intervertebral disc was resected, the bony endplate was prepared and no residual intervertebral disc was detected. Autologous bone particles were implanted in the intervertebral space and an intervertebral fusion cage with a decompressed autologous bone was implanted. Continue to decompress under the lamina and facet joints on both sides to fully expand the spinal canal and nerve root canal. The connecting rods were connected to the pedicle screws on both sides and fixed firmly. The position of the implant was confirmed by fluoroscopy again. If the dura mater was found to be torn during the operation, sutures were given to repair the dural tear. Flush the surgical area with plenty of saline. After complete hemostasis of the surgical area, a drainage tube was placed, and the incision was sutured layer by layer.

### 2.3. Postoperative Management

All patients underwent X-ray and 3D CT examinations on the first day after operation. The patients were instructed to wear rigid lumbar braces within 3 months after surgery. A clinical and radiographic follow-up was performed at 3 months, 6 months and 1 year after surgery, and annually thereafter.

### 2.4. Outcome Evaluation

The operation time, intraoperative bleeding and postoperative drainage volume, postoperative hospital stay and surgical complications were evaluated from querying the medical records. The Visual Analogue Scale (VAS) [18] for lower back and lower extremity pain was assessed before surgery, 1 week after surgery, 3 months after surgery and at the final follow-up. The Oswestry Disability Index (ODI) [19] was assessed before surgery and at the final follow-up. The modified MacNab criteria [20] were used to evaluate the efficacy at the last follow-up. The patients were evaluated for operative time, postoperative hospital stay, recurrence and surgical complications.

### 2.5. Statistical Methods

Statistical analysis was performed using the SPSS 21.0 software(IBM Corp., Armonk, NY, USA). The measurement data were expressed as mean ± standard deviation and repeated measures analysis of variance was used to compare each time point before and after surgery. The χ2 test was used in the analysis of the categorical variables. The inspection level α = 0.05.

## 3. Results

Both groups completed the operation successfully. The average follow-up time was 17.88 ± 5.36 m in the OLIF-PD group and 20.20 ± 7.86 m in the PLIF group. In the OLIF-PD group, the average operation time was 83.75 ± 21.84 min, the intraoperative blood loss was 122.50 ± 50.92 mL, the postoperative drainage volume was 83.75 ± 33.78 mL and the average postoperative hospital stay was 4.63 ± 1.06 d. In the PLIF group, the average operation time was 167.50 ± 17.95 min, the intraoperative blood loss was 341.00 ± 113.46 mL, the postoperative drainage volume was 209.00 ± 66.40 mL and the postoperative hospital stay was 6.90 ± 2.13 d. The average operation time, blood loss and postoperative hospital stay in the OLIF-PD group were significantly shorter than those in the PLIF group (*p* = 0.000). There was one case of nerve palsy in the OLIF-PD group, which was improved after two weeks of nutrient nerve and neuropathic drug treatment. In the PLIF group, two cases of dural sac injury, one case of lower limb paralysis and two cases of wound infection occurred. The dural sac injury was repaired during the operation, the deep fascia was sutured tightly and the catheter was extubated three days after the operation. The patients with paralysis of the lower limbs improved after one month of symptomatic treatment of the nutritional nerve. The superficial wound infection healed after an intensive dressing change (Table 2).

In the OLIF-PD group, the low back pain VAS score was 6.25 ± 1.17 before operation, 2.63 ± 0.74 at two days after operation and 0.63 ± 0.52 at the last follow-up. The VAS score of low back pain in the PLIF group was 6.50 ± 1.32 points before operation, 4.40 ± 0.82 points two days after operation and 1.20 ± 0.62 points at the last follow-up. There was no statistical difference in the preoperative low back pain between the two groups, but the low back pain in the OLIF + PD group was significantly lower than that in the PLIF group at each follow-up time point after surgery, with statistical significance. There was no significant difference in the leg pain scores between the two groups after surgery, three months after surgery and at the last follow-up. The ODI score of the OLIF-PD group decreased from 63.75 ± 8.51 points before operation to 29.25 ± 8.07 points at the last follow-up, and the ODI score of the PLIF group decreased from 64.20 ± 7.02 points before operation to 27.00 ± 4.42 points at the last follow-up. There was no statistical difference between the two groups (Table 3). The excellent and good rate of the modified MacNab standard at the last follow-up was 87.5% (7/8) in the OLIF-PD group and 70% (7/10) in the PLIF group. A typical case is shown in Figure 2.

## 4. Discussion

The surgical management of ASD after lumbar fusion is challenging [8,21,22]. The dissection of the scar around the nerve structure and the reconstruction of the stability of the lumbar spine are the difficulties of revision surgery [23,24]. The traditional revision surgery is a posterior open surgery. It needs to separate the scar at the posterior of the lumbar spine, remove the previous screws, increase the diameter of the screws and either re-implant the pedicle screws or change the screw channel to restore the stability of the spine [11]. Posterior revision surgery causes great damage to the paraspinal muscles and soft tissues, great trauma, heavy bleeding and a high risk of nerve damage [25]. OLIF is a minimally invasive lumbar fusion surgery with an anterolateral approach, which enters the intervertebral disc space between the aorta and the psoas major muscle on the anterolateral side of the intervertebral disc. OLIF has been well established for the treatment of degenerative diseases of the lumbar spine [14,15]. In recent years, many researchers have reported that OLIF surgery for ASD involves indirect decompression by enlarging the intervertebral space [26,27]. However, for disc herniation or spinal stenosis accompanied by significant spinal compression, indirect decompression is often insufficient, and direct posterior decompression is required. We have used OLIF combined with posterior channel decompression to treat these diseases and achieved good results.

Our results found that the OLIF-PD group showed better improvement in the low back pain VAS scores in the early postoperative period compared with the PLIF group. These indicate that the OLIF-PD technique only performs ipsilateral lamina fenestration and partial facetectomy, preserving most of the posterior anatomical structure of the lumbar spine, which can effectively avoid the extensive stripping of the paraspinal muscles and damage to the lamina, ligamentum flavum and articular process structures during posterior revision surgery. The last follow-up results showed that the ODI and leg pain VAS scores in the OLIF-PD group were similar to those in the PLIF group, and both groups showed better clinical improvement. It showed that the OLIF-PD could obtain satisfactory decompression and fusion effects, and the long-term effects were satisfactory. In our study, the operative time, blood loss and postoperative hospital stay were significantly lower in the OLIF-PD group than in the PLIF group. Jin C et al. [16] used OLIF to treat 12 cases of ASD and found that the OLIF group was superior to the PLIF group in terms of operation time, blood loss and hospital stay. In our cases of OLIF-PD, although posterior decompression surgery was added compared with other, OLIF alone, studies, the operative time and blood loss were still lower than those of the PLIF group and the postoperative hospital stay was significantly shorter than that of the PLIF group. These results indicated that OLIF-PD was more conducive to the rapid recovery of patients, while ensuring the decompression effect.

The operative time, intraoperative blood loss and postoperative drainage volume of the OLIF-PD group were significantly better than those of the PLIF group. Although the same surgeon operated on all of the patients, these differences between the two groups were related to the surgical approach and surgical trauma. The OLIF surgical exposure was faster, while avoiding extensive muscle stripping and injury. However, the PLIF surgery required more time to be spent on the separation of the scar tissue and the replacement of the previous internal fixation system. In the PLIF group, there were two cases of dural sac tear and two cases of superficial wound infection, which required longer observation and wound care for the patients, resulting in a significantly longer postoperative hospital stay in the PLIF group. These indicate that OLIF-PD surgery causes less trauma and is more conducive to the postoperative recovery of patients.

Another finding of this study was that OLIF-PD had significantly lower complications than PLIF, which is also an important advantage of OLIF [27,28]. The OLIF procedure accesses the intervertebral disc through a natural anatomical approach between the aorta and the psoas muscle, and therefore has a reduced incidence of lumbar plexus injury compared to extreme lateral interbody fusion [14,15]. Although posterior decompression was performed in this study, the scope of posterior surgical operation is relatively limited, and it was not necessary to operate in the scar area of the previous posterior surgery, and the previous internal fixation system did not need to be replaced, thus greatly reducing the risk of posterior decompression surgery. In the PLIF group, the two cases of superficial wound infection were healed after intensive dressing changes, but the length of hospital stay was prolonged, and the patient satisfaction was reduced. The OLIF-PD complication rates in our study were similar to other OLIF alone studies [26,27], and were significantly lower than those of PLIF. In the OLIF-PD cases, only one patient experienced transient nerve palsy, which improved after two weeks. This indicates that OLIF-PD has high surgical safety.

In the ASD after lumbar fusion, the pain caused by spondylolisthesis, segmental instability, mild intervertebral disc herniation, etc., can achieve satisfactory results through OLIF surgery alone, but for patients with huge disc herniation and spinal stenosis, fusion alone is often not enough. In the study by Jin C et al. [16], 8.3% of patients required posterior reoperation due to cage subsidence. Ohtori et al. [29] reported a reoperation rate of 9.5% for OLIF, whereas there were no cases in our study requiring revision surgery.

As our surgical approach required the direct decompression of the posterior lumbar spine, a suitable sized cage was selected for the anterior fusion to avoid the stress caused by the excessive distraction of the intervertebral space. At the same time, the anterior screw should be implanted as close to the vertebral endplate as possible to obtain a better holding force. We performed the direct decompression of the spinal canal to avoid the risk of indirect decompression of OLIF due to cage subsidence.

Yang Z et al. [30] reported that OLIF combined with endoscopic transforaminal discectomy in the treatment of upper and lower intervertebral disc herniation ASD had a satisfactory clinical effect. Their procedure required different forms of anesthesia, starting with endoscopic discectomy under local anesthesia and then OLIF under general anesthesia. The operation time of their study was 127.27 ± 21.49 min, which was slightly longer than our study. The patients included in this study had ASD with simple intervertebral disc herniation, and the endoscopic decompression surgery was difficult for patients with ossifying disease and spinal stenosis. In our study, the OLIF-PD operation was completed under general anesthesia and the posterior decompression was performed under a 20 mm channel, which could directly remove part of the articular process and lamina, reducing the surgical trauma and achieving higher decompression efficiency.

## 5. Conclusions

In conclusion, for ASD requiring direct decompression after posterior lumbar fusion, compared with traditional PLIF revision surgery, OLIF-PD has a similar clinical effect, but reduces the operation time, blood loss, operation time, hospital stay and complications. OLIF + PD may be an alternative revision strategy for ASD with spinal stenosis. However, this study has several limitations. First, this study is a retrospective study, and there may be bias in the selection of the surgical methods. Although the same surgeon performed both surgical procedures, there may have been differences in the surgical preference, and cases from other medical centers and other surgeons were not included. At the same time, this study only included ASD patients in one medical center, but the number of cases is too small, which is the biggest limitation of this study. The small number of patients and the short postoperative follow-up time made the current results lack sufficient credibility. In the future, multicenter, large sample, prospective, long-term follow-up studies should be implemented to provide more comprehensive data on the long-term complications and efficacy.

## Figures and Tables

**Figure 1 jpm-13-00368-f001:**
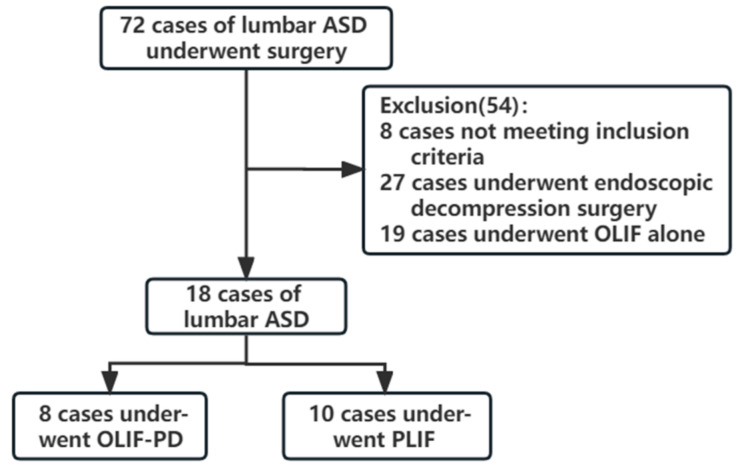
Flowchart of the experimental procedure.

**Figure 2 jpm-13-00368-f002:**
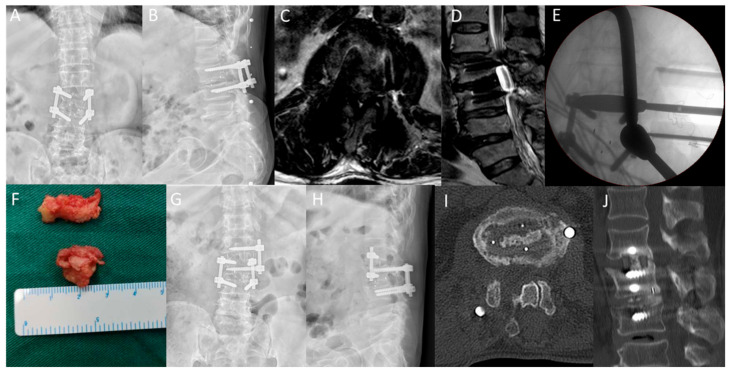
The patient, Male, 70 years old, was diagnosed L1/2 adjacent segment disease. (**A**,**B**): Anteroposterior and lateral lumbar X-rays suggest lumbar 2/3 internal fixation. (**C**,**D**): lumbar MRI revealed lumbar 1/2 disc herniation, significant compression of the dural sac and spinal canal stenosis. (**E**): C-arm shows the location of the fusion device during OLIF-PD surgery. (**F**): indicates disc tissue removed during posterior decompression surgery. (**G**,**H**): Postoperative lumbar anteroposterior and lateral X indicate good internal fixation position. (**I**,**J**): The 1-year follow-up indicated lumbar 1/2 interbody fusion, and the white arrow indicated the posterior bony excision area after channel decompression.

**Table 1 jpm-13-00368-t001:** Comparison of baseline data of two groups.

	OLIF-PD	PLIF	*p*
Age (y)	54.38 (12.48) 36–72	62.50 (9.77) 45–74	0.077
Sex (M/F)	5/3	5/5	0.556
BMI (kg/m^2^)	24.28 (1.90) 21.6–26.5	25.33 (1.29) 22.3–27.2	0.100
Previously fused levels			0.185
Single-level	4	7	
Double-level	2	3	
Triple-level	2	0	
Operated level			0.830
L1/2	1	1	
L2/3	3	2	
L3/4	3	4	
L4/5	1	3	
Follow-up time (months)	17.88 (5.36) 12–26	20.20 (7.86) 12–36	0.452

**Table 2 jpm-13-00368-t002:** Comparison of surgery-related data between the two groups.

	OLIF-PD	PLIF	*p*
Operation time (min)	83.75 (21.84) 60–120	167.50 (17.95) 140–200	0.00
Bleeding (ml)			
Intraoperative blood loss	122.50 (50.92) 50–200	341.00 (113.46) 150–500	0.00
Postoperative drainage	83.75 (33.78) 50–150	209.00 (66.40) 110–340	0.00
Postoperative hospital stay	4.63 (1.06) 3–6	6.90 (2.13) 5–11	0.00
Complications	1/8 (12.5%)	5/10 (50%)	0.049
Dural tear	0	2	
Nerve paralysis	1	1	
Wound infection	0	2	

**Table 3 jpm-13-00368-t003:** Comparison of VAS and ODI data between the two groups.

	OLIF-PD	PLIF	*p*
VAS of Low back			
Preoperative	6.25 (1.17) 5–8	6.50 (1.32) 4–8	0.644
2 days Postoperative	2.63 (0.74) 2–4	4.40 (0.82) 3–6	0.000
3 months Postoperative	1.75 (0.71) 1–3	2.50 (0.83) 1–3	0.033
Last follow-up	0.63 (0.52) 0–1	1.20 (0.62) 0–2	0.024
VAS of Leg			
Preoperative	5.50 (1.07) 4–7	5.70 (0.92) 4–7	0.624
2 days Postoperative	2.00 (0.53) 1–3	1.70 (0.66) 1–2	0.263
3 months Postoperative	1.25 (0.46) 1–2	1.20 (0.41) 1–2	0.781
Last follow-up	0.63 (0.52) 0–1	0.70 (0.66) 0–2	0.776
ODI			
Preoperative	63.75 (8.51) 54–72	64.20 (7.02) 58–74	0.886
Last follow-up	29.25 (8.07) 18–42	27.00 (4.42) 20–34	0.349

## Data Availability

The data presented in this study are available on request from the corresponding author.

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
