# Peer review of "Comparison of Oblique Lumbar Interbody Fusion Combined with Posterior Decompression (OLIF-PD) and Posterior Lumbar Interbody Fusion (PLIF) in the Treatment of Adjacent Segmental Disease(ASD)"

_jpm, 2023, doi:10.3390/jpm13020368_

Round 1

Reviewer 1 Report

One of the significant differences between the two groups is the operation time. So, it's relevant to mention the possible causes of these deferences, for example, the surgical teams' experience.

In addition, the postoperative stay of both groups is significantly extended. Could you explain that?

Author Response

Comments:One of the significant differences between the two groups is the operation time. So, it's relevant to mention the possible causes of these differences, for example, the surgical teams' experience.In addition, the postoperative stay of both groups is significantly extended. Could you explain that?

Response: Thank you for your advice. The difference in operative time between the two groups was mainly due to the difference in operative approach. Due to the revision surgery, the postoperative hospital stay of all patients in this study was longer than that of the primary operation, and the postoperative hospital stay of the PLIF group was longer, which was related to surgical complications. Those associated analyzes had been described in the discussion (Line226-236).

Reviewer 2 Report

Thank you very much for the opportunity to review the extremely interesting research. I appreciate the work, but I have a few comments:

1. Please describe the Study Design in detail

2. Please provide the age range of the patients (min-max), not just the average age

3. Please provide Study Limitation

4. In the Outcome Evaluation subsection, please provide references to the assessment scales used in the research

5. Please provide Flow Diagram.

Author Response

  1. Please describe the Study Design in detail

Response: Thank you for your advice. This study is a retrospective case-control study, and the detail information has been described in the Materials and methods (Line63-68).

Following IRB approval, we retrospectively reviewed 72 consecutively patients with lumbar ASD at our hospital, who underwent surgery for symptomatic ASD between December 2017 and April 2022. The same experienced spinal minimally invasive surgeon (corresponding author) performed all operations. Patients were screened according to the following inclusion and exclusion criteria. 

  1. Please provide the age range of the patients (min-max), not just the average age

Response: Thank you for your advice. The relevant data has been supplemented(table 1-3).

  1. Please provide Study Limitation

Response: Thank you for your advice. A Limitation paragraph had been added at the end of the manuscript (Line280-289).

      This study has several limitations. First, this study is a retrospective study, and there may be bias in the selection of surgical methods. Although the same surgeon performed both surgical procedures, there may have been differences in surgical preference, and cases from other medical centers and other surgeons were not included. At the same time, this study only included ASD patients in one medical center, but the number of cases is too small, which is the biggest limitation of this study. The small number of patients and the short postoperative follow-up time made the current results lack sufficient credibility. In the future, multicenter, large sample, prospective, long-term follow-up studies should be implemented to provide more comprehensive data on long-term complications and efficacy. 

  1. In the Outcome Evaluation subsection, please provide references to the assessment scales used in the research

Response: Thank you for your advice. Relevant references have been added (Reference18-20). 

  1. Please provide Flow Diagram.

Response: Thank you for your advice. Flow Diagram has been added (figure 1).

Reviewer 3 Report

Manuscript ID: jpm-2170913

Title: Comparison of Oblique Lumbar Interbody Fusion Combined with Posterior Decompression (OLIF-PD) and Posterior Lumbar Interbody Fusion (PLIF) in the Treatment of Adjacent Segmental Disease (ASD).

This study compared Oblique Lumbar Interbody Fusion Combined with Posterior Decompression (OLIF-PD) and Posterior Lumbar Interbody Fusion (PLIF) in the Treatment of Adjacent Segmental Disease (ASD). Some comments about this article:

1.     In this study, the number of subjects is small. Was a power analysis done about whether the number of subjects is sufficient?

2.     How about the functional outcome between these two surgical techniques?

3.   What are the limitations of this study?

4.     Table 2 compares “Wound infection” between OLIF-PD and PLIF approaches. Does the wound infection depend on the surgical approaches, but not post-operation wound care?

5.     How to obtain the average operation time and bleed loss? Are these data received from one surgeon? Is the surgeon representative of all surgeons?

Author Response

  1. In this study, the number of subjects is small. Was a power analysis done about whether the number of subjects is sufficient?

Response: Thank you for your comment. This study is a retrospective study. Although all ASD cases in our center in the past 5 years were included, the number of cases is too small, which is the biggest limitation of this study. Since there are very few ASD requiring surgery in clinical work, the number of patients with surgically treated ASD reported in the literature were also small (26,44,65 Reference1-3). Our study focuses on providing a novel surgical revision strategy (OLIF-PD) for ASD. In the future, prospective, multicenter studies may include more patients to provide more accurate statistical analysis.

Reference 1:Louie PK, Varthi AG, Narain AS, et al. Stand-alone lateral lumbar interbody fusion for the treatment of symptomatic adjacent segment degeneration following previous lumbar fusion. Spine J. 2018;18(11):2025-2032. doi:10.1016/j.spinee.2018.04.008

Reference2:Screven R, Pressman E, Rao G, Freeman TB, Alikhani P. The Safety and Efficacy of Stand-Alone Lateral Lumbar Interbody Fusion for Adjacent Segment Disease in a Cohort of 44 Patients. World Neurosurg. 2021;149:e225-e230. doi:10.1016/j.wneu.2021.02.046

Reference 3:Li T, Zhu B, Liu X. Revision Strategy of Symptomatic Lumbar Adjacent Segment Degeneration: Full Endoscopic Decompression versus Extended Posterior Interbody Fusion. World Neurosurg. 2020;142:e215-e222. doi:10.1016/j.wneu.2020.06.168 

  1. How about the functional outcome between these two surgical techniques?

   Response: Thank you for your comment. In terms of functional outcomes, this study compared the differences in ODI score and modified MacNab standard between the two surgical techniques, and the relevant data have been described in the results(Line172-177,207-224). 

  1. What are the limitations of this study?

Response: Thank you for your advice. A Limitation paragraph had been added at the end of the manuscript(Line280-289).

This study has several limitations. First, this study is a retrospective study, and there may be bias in the selection of surgical methods. Although both surgical procedures were performed by the same surgeon, there may have been differences in surgical preference, and cases from other medical centers and other surgeons were not included. At the same time, this study only included ASD patients in one medical center, but the number of cases is too small, which is the biggest limitation of this study. The small number of patients and the short postoperative follow-up time made the current results lack sufficient credibility. In the future, multicenter, large sample, prospective, long-term follow-up studies should be implemented to provide more comprehensive data on long-term complications and efficacy. 

  1. Table 2 compares “Wound infection” between OLIF-PD and PLIF approaches. Does the wound infection depend on the surgical approaches, but not post-operation wound care?

Response: Thank you for your advice. In terms of wound infection complications, the PLIF group was significantly higher than the OLIF-PD group. We analyzed that the difference between the two groups mainly came from the surgical approach. The OLIF-PD group did not need to pass through the original surgical area, and the damage was smaller. These may be the main cause of wound infection. Although 2 cases of superficial wound infection in the PLIF group were healed after dressing change and wound care, the length of hospital stay was prolonged and patient satisfaction was reduced. The related analysis has been added to the Discussion (Line237-250). 

  1. How to obtain the average operation time and bleed loss? Are these data received from one surgeon? Is the surgeon representative of all surgeons?

Response: Thank you for your advice. The operation time and blood loss were obtained by querying medical records, and the blood loss included intraoperative blood loss and postoperative drainage. Although the same experienced surgeon performed all operations in this study, cases from other medical centers and other doctors were not included. One surgeon cannot represent all other surgeons, and there may be differences in surgical operation preferences. This is also one of the limitations of this study, which has been stated in the manuscript, and a multi-center study should be implemented later.

Round 2

Reviewer 3 Report

All comments have been answered and revised, this paper can be published.